# Size-resolved condensation sink as an approach to understand pathways how gaseous emissions affect health and climate

Teemu Lepistö<sup>1</sup>, Hilkka Timonen<sup>1,2</sup>, Topi Rönkkö<sup>1</sup>, Miikka Dal Maso<sup>1</sup>

<sup>1</sup>Aerosol Physics Laboratory, Tampere University, Tampere, 33720, Finland <sup>2</sup>Atmospheric Composition Research, Finnish Meteorological Institute, Helsinki, 00101, Finland

Correspondence to: Teemu Lepistö (teemu.lepisto@tuni.fi)

Abstract. Vapour condensation onto existing aerosol particles is important regarding aerosol health and climate effects. Existing particles can act as carriers for toxic vapours into the human respiratory tract. Also, condensation changes the aerosol optical properties. Condensation sink (CS) is a widely utilised parameter in atmospheric aerosol studies that estimates the attachment rate of vapour molecules onto existing particles. However, typically only the total CS is investigated. Here, we explore the concept of size-resolved condensation sink (CS size distribution). We calibrate an electrical low pressure impactor to measure CS and then utilise the method in urban aerosol measurements conducted in Finland, Germany, Czechia and India, covering road traffic sites, airports, detached housing residential areas, industrial and shipping sites. We report considerably varying shapes and mean sizes of CS size distributions: CS in Finland was more attributable to ultrafine particles (geometric mean diameters being 85–206 nm) than in Central Europe (151–263 nm) and India (278 nm). We introduce a parameter CS attributable to ultrafine particles (CS<sub>0.1</sub>), which may be especially relevant when considering the formation of cloud condensation nuclei as well as deposition of condensed vapours in the human lung. Furthermore, the results show that the formation and changes of the atmospheric particle size distribution vary in different conditions and environments. Thus, adaptation of CS size distribution could be a simple but effective tool to consider these differences in climate models. Overall, CS size distribution can improve general understanding of the effects of gaseous emissions on health and climate.

#### 1 Introduction




Condensation of vapour molecules onto existing particles is a key aerosol aging process in terms of health and climate effects of ambient aerosols. Existing aerosol particles can contribute to health impact mechanisms as carriers of condensed and potentially toxic compounds into the human respiratory tract (e.g. Ali et al. 2020). For example, a toxicological study by Hakkarainen et al. (2022) indicates that organic coating on soot particles increases the aerosol toxicity. In the atmosphere, condensation of initially gaseous compounds into the particulate phase is an important process in the growth of secondary aerosol particles which contribute a significant fraction of ambient PM<sub>2.5</sub> mass around the world (e.g., Chen et al. 2022, Mishra et al. 2023). In general, elevated concentrations of ambient PM<sub>2.5</sub> have been linked with adverse health impacts which means that also the condensation processes in PM<sub>2.5</sub> formation are relevant from health impact point of view. Furthermore, secondary organic aerosol from anthropogenic sources has been suggested to be one of the most important contributors on aerosol oxidative potential in most parts of Europe (Daellenbach et al. 2020).

Climate-wise, condensation of atmospheric vapours directly affects aerosol optical properties and, thus, on how the particles interact with light. For example, light absorption of soot particles can be increased due to soot particle coating with organic and inorganic compounds, causing so-called lensing effect (Riemer et al. 2019). On the other hand, particles larger than 50–100 nm can act as cloud condensation nuclei (CCN) (Kerminen et al. 2012) contributing to cloud formation and, thus, to the net cooling effects of clouds (e.g. Fuzzi et al. 2015). Typically, recently formed particles, e.g., in exhaust (Rönkkö et al. 2017) or during new particle formation (NPF) events (Kontkanen et al. 2017), are mostly in the sizes below 10 nm, hence, condensation processes are needed to increase the particles size so that they can act as CCN.

In addition to direct health and climate effects, vapour condensation on existing particles is important to understand also in terms of new particle formation processes (Pirjola et al. 1999; Zhang et al., 2004). The condensation rate of vapour molecules onto particles depends on the existing particle concentration which is commonly estimated with a parameter called condensation sink (CS) (Pirjola et al. 1999). It has been suggested that high CS of existing particles indicates that vapour molecules in the air mainly condensate onto the existing particles. Low CS, on the other hand, may result in NPF-events as there is not high enough concentration of existing aerosol particles for all vapour molecules to condensate, causing more likely nucleation. Many studies have reported that stronger NPF events tend to occur during low CS periods (e.g., Boy and Kulmala 2002, Hamed et al. 2007, Zaidan et al. 2018). Therefore, CS can be a highly useful parameter in estimating and modelling the effects aerosol aging on the air quality and climate.

In atmospheric aerosol studies, typically only the total CS of particles is utilised as a parameter. However, both the health and climate effects of aerosol particles are strongly dependent on the particle size, indicating that also the effects related to condensation are strongly affected by the size of the particles the vapour molecules are condensed onto. A common approach to measure total CS is a number size distribution measurement e.g., with scanning or differential mobility particle sizers (SMPS/DMPS) (e.g., Hamed et al. 2007, Dal Maso et al. 2008, Zaidan et al. 2018). However, with this method, the detailed shape information (and, thus, surface area) cannot be accurately considered, causing potentially uncertainty in the CS

measurement. Another way to measure total CS is the utilisation of a diffusion charging based instrument (Kuuluvainen et al. 2010) where the number of elementary charges particles carry after diffusion charging is related to particle surface area, similarly as CS. Thus, electric current caused by the measured diffusion charged particles should in principle correlate well with CS. Diffusion charger -based measurement for CS, however, has been previously utilised only in terms of total CS measurement.

Here we explore the condensation sink (CS) size distribution concept and investigate the benefits of size-resolved CS measurement for aerosol studies. We aim to expand the diffusion charger -based measurement to be suitable for CS size distribution by reporting the conversion factors that can be used to convert the raw data measured with the electrical low pressure impactor (ELPI+) instrument to CS. After that, this CS calibration of ELPI+ is utilised with ambient aerosol measurement data collected in various urban environments in Finland, Germany, Czechia and India, including road traffic sites, detached-housing areas, airports, industrial and shipping sites. Furthermore, we compare CS size distribution characteristics with particle number and mass concentration as well as estimate geometric mean diameters of the CS size distributions (together with CS diameter). Moreover, we investigate the differences of CS size distribution measurement between a mobility particle sizer (DMPS) and the ELPI+.

## 2 Experimental

90

# 0 2.1 Electrical low pressure impactor

The ELPI+ (Keskinen et al. 1992) measures particle size distributions by utilising diffusion charging and a cascade impactor. First, the sampled particles are charged in a diffusion charger. According to Järvinen et al. (2014), the ELPI+ charging efficiency (Pn) as a function of the mobility equivalent diameter is

$$Pn = \begin{cases} 68.531 d_{\rm p}^{1.225}, & d_{\rm p} < 1.035 \; \mu{\rm m} \\ 67.833 d_{\rm p}^{1.515} & 1.035 \; \mu{\rm m} \leq d_{\rm p} \leq 4.282 \; \mu{\rm m}, \\ 126.83 d_{\rm p}^{1.085}, & d_{\rm p} > 4.282 \; \mu{\rm m} \end{cases} \tag{1}$$

where P is the particle penetration through the charger and n is the number of elementary charges carried by particles after charging. The detected electric current on the impactor stages is a multiplication of Pn, elementary charge (e) and sample flow (Q). The nominal sample flow of the ELPI+ is 10 lpm. After charging, particles are classified according to their aerodynamic size in a 14-staged cascade impactor, enabling size distribution measurement of the diffusion charged current caused by the collected particles with 1 s time-resolution in the size range of 6 nm - 10  $\mu$ m. As the electric current caused by the particles is known as a function of particle size, the electric current data can be converted into particle metrics like number size distribution. As the electric current caused by the particles depends on the mobility equivalent size of particles, and the size classification depends on the aerodynamic size, the particle effective density ( $\rho_{\rm eff}$ ) needs to be determined for an accurate measurement. In this study, all the reported parameters, i.e., CS, PM<sub>2.5</sub> and particle number (PN) concentrations are based on the ELPI+ data of the same unit.

120

95 The classic ELPI has been calibrated to measure total CS before by Kuuluvainen et al. (2010). Diffusion charging (where ions collide with sampled particles) is related to natural condensation (where vapor molecules collide with existing particles). Thus, conversion of diffusion charged current into CS is a suitable method for CS measurement. Kuuluvainen et al. (2010), however, only calibrated the conversion from total electric current measured from all the impactor stage into total CS, not enabling measurement of size-resolved CS. This conversion is referred here as a single-factor calibration. Also, the renewed ELPI+ has not been utilised in CS measurement earlier according to our knowledge.

## 2.2 Condensation sink calibration for the electrical low pressure impactor

CS (unit 1/s) is calculated as a multiplication of the particle number concentration and the attachment rate factor of vapour molecules onto the particles. The attachment rate factor ( $A_{CS}$ ) can be calculated with an equation

$$A_{CS} = 2\pi d_p D\beta, \tag{2}$$

where  $d_p$  is the particle diameter, D is diffusion constant for the considered vapour molecule in air and  $\beta$  the Fuchs-Sutugin correction factor (Fuchs and Sutugin 1971). According to Poling et al. (2010), D is

$$D = 0.00143 \frac{T^{1.75} \sqrt{M_{\text{air}}^{-1} + M_{\text{x}}^{-1}}}{P\left(D_{\text{x,air}}^{\frac{1}{3}} + D_{\text{x,vap}}^{\frac{1}{3}}\right)^{2}},$$
(3)

where  $D_x$  is the diffusion volume, P ambient pressure and M molecular weight (Poling et al. 2000). For air, molecular mass and diffusion volume were set as 28.965 g/mol and 19.7, respectively. For sulfuric acid, the same values were 98.08 g/mol and 51.66 (Poling et al. 2000). The Fuchs-Sutugin correction is calculated with a formula

$$\beta = \frac{1 + \text{Kn}}{1 + \left(\frac{4}{3\alpha} + 0.377\right) \text{Kn} + \frac{4}{3\alpha} \text{Kn}^2},\tag{4}$$

where  $\alpha$  is the mass accommodation coefficient and Kn is the Knudsen number, i.e., the relationship between the particle diameter ( $d_p$ ) and the mean free path of the condensing vapour molecules ( $\lambda_{\text{vap}}$ ):

$$Kn = \frac{2\lambda_{\text{vap}}}{d_{\text{p}}}.$$
 (5)

Here, the particle mobility equivalent diameter was used in the calculation of the Knudsen number. The mean free path of the vapor molecules depends on the mass of a single vapour molecule ( $m_{\text{vap}}$ ), and ambient temperature (T):

$$\lambda_{\text{vap}} = 3D \sqrt{\frac{\pi m_{\text{vap}}}{8k_{\text{h}}T}} \,. \tag{6}$$

The mass accommodation coefficient was assumed to be 1.0. Previous studies have not found consensus value on the mass accommodation coefficient (e.g., Pöschl, et al. 1998, Hanson 2005), but it has been suggested to be dependent e.g., on the temperature and composition of existing particles (Roy et al. 2020).

Now, to calibrate the ELPI+ to measure size-resolved CS, the charging efficiency of the diffusion charger needs to be considered (Equation 1). The size dependent CS response function (*K*) for ELPI+ is



$$K = \frac{A_{CS}}{PneQ} = \frac{2\pi d_p D\beta}{PneQ},\tag{7}$$

which is shown in Fig. 1 as a function of particle size, together with the ELPI+ charging efficiency and attachment rate factor. For the CS size distribution measurement, the conversion factor from the electric current into CS can be determined for each impactor stage of the ELPI+ based on the response function and the average size of collected particles onto each stage ( $d_{p,mean}$ ). These conversion factors corresponding to the unit calibrated by Järvinen et al. (2014) are collected in Table 1. This calibration method is referred here as the stage-specific calibration. In addition, the single factor calibration for the ELPI+ was done as it gives valuable information of the performance of diffusion charger -based measurement of CS in cases where particle size is not known (like electrical particle sensors). The single-factor conversion from total measurement electric current to total CS was based on the value of K at 150 nm, being 23.9\*10<sup>-6</sup> (1/(sfA)).

It should be noted that the response function depends on the chosen  $\rho_{\rm eff}$  as described in Section 2.1. As the effective density cannot accurately be determined only with ELPI+ measurement, assumptions or additional instruments are needed, which can be considered as a downside of the method. The effect of  $\rho_{\rm eff}$  on the K is shown in Fig. 1. On the other hand, as mentioned in Section 2.1., the diffusion charger -based measurement likely represents CS better than particle number size distribution measurement e.g., with DMPS or SMPS, where particles typically need to be assumed to be spherical.

Figure 1: a) ELPI+ charger efficiency (*Pn*) and attachment rate factor (*A*<sub>CS</sub>) as a function of particle mobility equivalent diameter. b) ELPI+ CS response coefficient K as a function of particle aerodynamic diameter with different particle effective densities.

Table 1: ELPI+ CS conversion factors from electric current to CS for different stages with four different particle effective densities for the unit used in this study.

ELPI+ CS conversion factor  $\times$  10<sup>-6</sup> (1/(sfA))

|              |                                     |                                     | ` ` '/                              |                                     |  |
|--------------|-------------------------------------|-------------------------------------|-------------------------------------|-------------------------------------|--|
| Stage        | $\rho_{\rm eff} = 1.0 \ \rm g/cm^3$ | $\rho_{\rm eff} = 0.8 \ \rm g/cm^3$ | $\rho_{\rm eff} = 1.5 \ \rm g/cm^3$ | $\rho_{\rm eff} = 1.8 \ \rm g/cm^3$ |  |
| (dpMean)     |                                     |                                     |                                     |                                     |  |
| 1 (9.71 nm)  | 2.99                                | 3.54                                | 2.19                                | 1.90                                |  |
| 2 (21.8 nm)  | 5.54                                | 6.56                                | 4.08                                | 3.55                                |  |
| 3 (40.6 nm)  | 8.73                                | 10.7                                | 6.46                                | 5.63                                |  |
| 4 (71.4 nm)  | 13.1                                | 15.3                                | 9.81                                | 8.59                                |  |
| 5 (121 nm)   | 18.5                                | 21.2                                | 14.2                                | 12.5                                |  |
| 6 (198 nm)   | 24.8                                | 27.7                                | 19.6                                | 17.5                                |  |
| 7 (311 nm)   | 30.6                                | 33.1                                | 25.4                                | 23.0                                |  |
| 8 (478 nm)   | 35.2                                | 37.0                                | 30.8                                | 28.5                                |  |
| 9 (752 nm)   | 38.7                                | 39.1                                | 35.6                                | 33.9                                |  |
| 10 (1.24 μm) | 37.4                                | 34.9                                | 38.7                                | 37.9                                |  |
| 11 (2.00 μm) | 31.8                                | 29.1                                | 36.6                                | 38.6                                |  |
| 12 (2.99 μm) | 27.0                                | 24.5                                | 31.8                                | 34.0                                |  |
| 13 (4.41 μm) | 23.1                                | 23.0                                | 27.2                                | 29.4                                |  |
| 14 (7.27 μm) | 22.7                                | 22.4                                | 23.1                                | 23.7                                |  |
|              |                                     |                                     |                                     |                                     |  |

In addition, it needs to be noted that the attachment rate factor and, thus, the CS response function, depends on ambient conditions, like temperature and pressure (Equations 2 and 5). Here we assumed the normal temperature and pressure (NTP) conditions (T = 20°C, P = 1 atm). K with different ambient temperature and pressure values are shown in Fig. S1. Also, the mentioned uncertainties related to the mass accommodation coefficient should be acknowledged.

## 2.2.1 Validation of the calibration

First, to understand the theoretical performance of the ELPI+ CS measurement, nine different particle number size distributions (Fig. 2) were simulated according to the ELPI+ collection efficiency functions (Järvinen et al. 2014). Then, the simulated ELPI+ CS results were compared to the theoretical one of each simulated distribution, calculated based on Equation 2. The simulated distributions were based on results reported by Sebastian et al. (2022), Teinilä et al. (2022) and Trechera et al. (2023), covering various environments and conditions in Europe and India with varying regional pollution levels. The distributions were re-created by utilising log-normal size distributions to roughly match the reported particle number size distributions. Also, ρ<sub>eff</sub> was set to 1.0 g/cm<sup>3</sup>. With the simulations, the uncertainty of the stage-specific





calibration in the CS conversion due to the reduced size resolution of the 14-staged impactor measurement can be tested. In addition, the accuracy of the single-factor CS calibration compared to the stage-specific one was tested with the simulated size distributions.

Figure 2: Simulated particle number size distributions. Traffic (HEL), WB (HEL) and LRT (HEL) indicate periods contributed by road traffic, residential wood burning and a long-range transported episode in measurements by Teinilä et al. (2022) in Helsinki. Urban background (UB) data from Helsinki and Budapest (BUD), as well as traffic data from Dresden (DRE) and rural background (BG) data from Po Valley (POV) are based on study by Trechera et al. (2023). Urban background data from Delhi (DEL) and Hyderabad (HYD) are based on study by Sebastian et al. (2022).

Second, to evaluate the effect of particle  $\rho_{\rm eff}$  on the CS measurement, CS measurement of an ELPI+ and a DMPS were compared based on data measured in a street canyon in Helsinki in winter 2022 (Lepistö et al. 2024, Teinilä et al. 2025). The CS from the DMPS data was calculated with the same approach as for the ELPI+ (Equation 2). The data were divided into three categories based on conditions: 1. low regional background concentration, 2. temperature inversion, 3. long range transported (LRT) episode. Average concentrations of these periods are collected in the supplementary information (Table S1). Most importantly, the average  $\rho_{\rm eff}$  was different during the periods, and the ELPI+ CS measurement was compared to DMPS by assuming unit density but also by correcting the measurement by using the estimated average  $\rho_{\rm eff}$  of the periods based on Equation 8. The average  $\rho_{\rm eff}$  was estimated by comparing the peak sizes of particle surface area size distributions of the DMPS and ELPI+ (see Lepistö et al. 2024). In the analysis, all the particles were assumed to have the same  $\rho_{\rm eff}$  even though, in reality,  $\rho_{\rm eff}$  has size-dependent and temporal variation. However, the cascade impactor measurement of the ELPI+ fundamentally challenges the use of size-dependent values for  $\rho_{\rm eff}$ . In addition, the measured ELPI+ data in different countries and locations (Section 2.3) were converted into CS by using varying  $\rho_{\rm eff}$  values from 0.8 g/cm³ to 1.5 g/cm³ to see how  $\rho_{\rm eff}$  changes the measured total CS and the size distribution. According to many studies,  $\rho_{\rm eff}$  of ultrafine particles is typically near 1.0 g/cm³ whereas for larger accumulation mode particles  $\rho_{\rm eff}$  can be around 1.5–1.8 g/cm³ (e.g., Levy et al.


2013, Yin et al. 2015, Lu et al. 2024). The relationship between the particle mobility equivalent diameter ( $d_{\rm m}$ ) and the aerodynamic diameter ( $d_{\rm a}$ ) can be calculated from

$$d_{\rm m} = d_{\rm a} \sqrt{\frac{C_{\rm c}(d_{\rm a})}{\rho_{\rm eff} C_{\rm c}(d_{\rm m})}},\tag{8}$$

where  $C_c$  is the Cunningham slip correction factor. Also, the size distribution results of the DMPS were converted from mobility equivalent diameter to aerodynamic (Fig. 4) by utilising Equation 8.

## 2.3 Measurement campaigns

Our study utilises data from eight measurement campaigns conducted in Helsinki (Finland), Tampere (Finland), Raahe (Finland), Düsseldorf (Germany), Prague (Czechia) and Delhi-NCR (India) in 2018–2022. These campaigns are listed in Table 2, and brief descriptions of the campaigns are provided in Table S2. Data from seven of the campaigns have been utilised in a study where particle lung deposited surface area (LDSA<sup>al</sup>) concentrations and size distributions were compared by Lepistö et al. (2023), and the same dataset is utilised also in this study. In addition, data from the measurement campaign conducted in Helsinki during winter 2022 (Lepistö et al. 2024, Teinilä et al. 2025) were utilised in the comparison of DMPS and ELPI+ CS measurement.

Table 2: Measurement campaigns included in this study. See campaign descriptions in Table S2.

| Time                           | Microenvironment                                                                                                                                                        |
|--------------------------------|-------------------------------------------------------------------------------------------------------------------------------------------------------------------------|
| 18 January – 16 February 2022  | Urban traffic                                                                                                                                                           |
|                                |                                                                                                                                                                         |
| 13 – 23 August 2019            | Urban traffic, Highway,                                                                                                                                                 |
|                                | Harbour                                                                                                                                                                 |
| 1 – 11 March 2021              | Urban traffic, Airport,                                                                                                                                                 |
|                                | Residential area                                                                                                                                                        |
| 29 April – 14 May 2020         | Highway                                                                                                                                                                 |
| 25 January – 4 February 2021   | Residential area, industrial                                                                                                                                            |
| 8 – 23 March 2022              | Urban traffic, Highway,                                                                                                                                                 |
|                                | Airport, Riverside                                                                                                                                                      |
| 25 March – 3 April 2022        | Urban traffic, Highway                                                                                                                                                  |
| 16 November – 14 December 2018 | Urban traffic                                                                                                                                                           |
|                                | 18 January – 16 February 2022  13 – 23 August 2019  1 – 11 March 2021  29 April – 14 May 2020  25 January – 4 February 2021  8 – 23 March 2022  25 March – 3 April 2022 |





Briefly, the comparison of CS measurement between the DMPS and ELPI+ was done in a street canyon located in the city centre of Helsinki (60.1963° N, 24.9523° E) on 18 January–16 February 2022. During the measurements, contribution of regional aerosol was mainly low, but On 31 January–5 February an inversion episode, and on 13–16 February a long-range transported (LRT) pollution episode affected the measured aerosol, e.g., in terms of effective density and average particle size, enabling comparison of the methods with different aerosol characteristics. All the other campaigns were conducted next to a certain urban aerosol source (road traffic, shipping, airport, detached housing residential area or industrial area). The focus of the campaigns was to study the characteristic aerosol emissions of the studied sources. Hence, the measurements were targeted to be conducted when the measurement sites were clearly affected by the targeted emission source. For example, near the airports, only downwind periods are considered in the results (see Table S2).

#### 3 Results

# 3.1 Validation of the condensation sink calibration for the electrical low pressure impactor

The simulated total CS values with both the stage-specific and single-factor methods are compared to the theoretical ones in Fig. 3. The simulated stage-specific total CS was 3.5–4.2 % lower than the theoretical value, showing that the 14-impactor-stage-measurement of the ELPI+ should be good enough to measure CS accurately regardless of the particle size distribution. Also, the simulated CS size distributions were similar compared to the theoretical values (Fig. S2-4). With the single-factor calibration, the simulated CS was -18.2–29.0 % of the theoretical value, but with 7 of the 9 simulated distribution the difference was less than  $\pm$  12 %, supporting the results by Kuuluvainen et al. (2010) showing that the diffusion charger -based measurement even without particle size analysis is an effective method for indicative CS measurement. As the single-factor CS calibration factor was chosen based on size 150 nm, the single-factor method overestimates CS of size distributions having high concentrations of ultrafine particles, whereas it underestimates size distributions with high concentrations of > 200 nm particles (see Fig. 1).



Figure 3: Simulated total CS of the size distributions in Fig. 2 with both stage-specific and single-factor calibrations compared to the theoretical value. T indicates traffic site, whereas other abbreviations are the same as in Fig. 2. The shaded lines represent error limits of ± 20 %.

The average CS size distributions measured with both the ELPI+ and DMPS in Helsinki during the low background, inversion and LRT periods of the comparison campaign (in 2022, Table 2) are shown in Fig. 4. In principle, both instruments measured rather similarly shaped size distributions. However, the ELPI+ reports higher total CS during all the periods compared to the DMPS. The (density corrected) ELPI+ measured 1.31, 1.25 and 1.20 times higher total CS than the DMPS during the low background, inversion and LRT episodes, respectively. The difference is most likely related to the fractal structure of particles, which can be better observed with the electric current measurement by the ELPI+. The difference between the devices decreased as the average size of CS size distribution and average effective density increased. The larger accumulation mode particles represent aged aerosol, for which it can be considered that the particles are less agglomerated (e.g., Rissler et al. 2014), hence, decreasing the difference between the instruments. Also, it can be seen that the relative difference between the instruments was clearly the highest with particles around 100 nm or smaller, which typically represent rather fresh nearby emissions, typically having more agglomerated structures of particles (e.g., Rissler et al. 2014). The unit-density assumed ELPI+ measured 1.05, 1.15 and 1.32 times higher total CS compared to the density corrected measurement, respectively. Therefore, in terms of total CS measurement, it is important to consider the average effective density with the ELPI+ measurement, especially if the concentration of accumulation mode particles is high, as they typically have higher effective density.



Figure 4: Comparison of average ELPI+ and DMPS CS size distributions measured during low background (Low BG), inversion and long range transported (LRT) pollution periods in the ELPI+ and DMPS comparison campaign in Helsinki in winter 2022. The ELPI+ results were calculated by assuming the unit effective density ( $\rho_{eff} = 1.0 \text{ g/cm}^3$ ) and by considering the estimated average effective density ( $\rho_{eff}$  corr.).

On the other hand, the effective density does not considerably affect the shape of the CS size distribution, showing that both ELPI+ methods are especially suitable to measure the size-resolved CS, even if the effective density of particles cannot be determined. In Section 3.2., the size-resolved CS from the other studied sites are presented and compared (Fig. 5). In Fig. S5-7, the same results are shown with different values of  $\rho_{\text{eff}}$  (0.8 and 1.5 g/cm<sup>3</sup>) utilised in the calculation. Similar to Fig. 4, the effective density did not considerably affect the shape and average size of the distribution, but the differences in the average total CS varied from 0.77 to 1.14 (Fig. S8). Thus, regarding the suitability of ELPI+ in terms of CS measurement, it can be concluded that size-resolved measurement is accurate even without considering the  $\rho_{\text{eff}}$ , but in terms of total CS, the result can vary roughly  $\pm 30$  % if the  $\rho_{\text{eff}}$  cannot be accurately estimated. Also, the mobility size distribution -based CS measurement seems to measure roughly 20–30 % lower CS than the diffusion-charger based measurement (Fig. 4), but it should be noted that this ratio can be different in other environments.

## 3.2 Condensation sink in different urban environments

In Fig. 5, the average CS size distributions measured at all the studied environments (except the ELPI+ and DMPS comparison campaign) are shown. In Table 3, geometric mean diameters (GMD<sub>CS</sub>) of the CS size distributions together with CS diameter (CSD, Lehtinen et al. 2003), as well as total CS, PM<sub>2.5</sub> and PN are collected. In principle (in case of a lognormal size distribution), the GMD<sub>CS</sub> represents the diameter of a particle that roughly 50 % of CS is contributed by particles smaller and larger than the GMD<sub>CS</sub> size. The CS diameter, on the other hand, represents the diameter of monodisperse aerosol particles that would contribute to equal total CS if the total number of the monodisperse particles was the same as the






measured PN concentration. In Fig. 5, CS size distributions clearly varied depending on the environment. In locations, where ultrafine particle concentrations were high (road traffic sites, airport), GMD<sub>CS</sub> sizes were considerably smaller compared to other sites, being below 100 nm near airport and road traffic in Finland. GMD<sub>CS</sub> of below 100 nm indicates that vapour molecules condense onto ultrafine particles more likely than on accumulation mode particles (larger than 100 nm), which may be important e.g., in terms of particle lung deposition or in the formation of CCN nuclei. As PM<sub>2.5</sub> concentrations increased, also the CS size distributions shifted to larger sizes in terms of both GMD<sub>CS</sub> and CSD values. In Finland, GMD<sub>CS</sub> sizes were 85–206 nm, whereas in Central Europe and India, 152–263 nm and 278 nm, respectively. Mostly, GMD<sub>CS</sub> sizes were between 150–300 nm. Therefore, increased PM<sub>2.5</sub> concentration relatively decreased the condensation on ultrafine particles.

Overall, according to the results, in locations with low PM<sub>2.5</sub> and high ultrafine particle concentration, CS can considerably be contributed by particles smaller than 100 nm, whereas in locations with high PM2.5, the average size of CS size distribution seems to reach a plateau size around 300 nm. It should be taken into account that the reported diameters represent the aerodynamic size of particles. For example, if assuming  $\rho_{\rm eff}$  to be 1.7 g/cm<sup>3</sup>, the 150–300 nm range would be 100-213 nm in mobility equivalent size (Eq. 8), matching well the typical median size of accumulation mode particles (Rose et al. 2021, Leinonen et al. 2022). Therefore, the CS distributions also explain why the accumulation mode of particles is typically always seen in this size range regardless of the environment, as the GMD<sub>CS</sub> size does not significantly increase compared the moderately polluted Central Europe (PM<sub>2.5</sub>: 17.8–28.6 µg/m³) and highly-polluted India (PM<sub>2.5</sub>: 256.9 µg/m³). The relationship between GMD<sub>CS</sub> size and the more commonly utilised CSD size, was rather constant, the CSD size being 26-46 % of the GMD<sub>CS</sub> size (average 35 %). It should be noted that the two parameters have radically different interpretations and uses: the CS diameter is useful when trying to simplify the situation while conserving both CS and PN (and the size in which the number of growing particles is largest), while GMD<sub>CS</sub> gives an estimate of the size range where the majority of CS is and where the number of condensing molecules end up. Literature data on both parameters is scarce; however, in comparison to CSD numbers reported by Dal Maso et al. (2008), we observe CSDs at lower sizes. This is expected as the numbers by Dal Maso et al. are from boreal background stations representing mostly clean air, while our study reports comparatively strong influence of nearby aerosol sources.



Figure 5: The average CS size distributions in the studied environments. The dashed line represents the size of 100 nm. Note varying y-axis. H and T indicate Helsinki and Tampere, respectively, whereas SC and HW indicate, street canyon and highway. Urban indicates an urban traffic site and ResArea a residential area.

Table 3: The average total CS in the studied environments as well as the geometric mean diameter (GMD<sub>CS</sub>) and condensation sink diameters (CSD, Lehtinen et al. 2003) of CS size distributions. Also, average PM<sub>2.5</sub> and PN concentrations are shown.

|          |                       |                         | are shown.   |                           |                       |                         |
|----------|-----------------------|-------------------------|--------------|---------------------------|-----------------------|-------------------------|
| Country/ | Location              | $\boldsymbol{GMD}_{CS}$ | CSD          | CS*10 <sup>-2</sup> (1/s) | PM2.5 ( $\mu g/m^3$ ) | PN (1/cm <sup>3</sup> ) |
| Region   |                       | (nm)                    | (nm)         |                           |                       |                         |
| Finland  | Helsinki: Airport     | 85.0                    | 27.5         | 0.73                      | 5.0                   | 50 100                  |
|          | Helsinki: Street      | 92.5                    | 31.4         | 0.35                      | 3.2                   | 18 300                  |
|          | canyon (Winter)       |                         |              |                           |                       |                         |
|          | Tampere: Highway      | 98.1                    | <i>35.7</i>  | 1.18                      | 12.3                  | 48 500                  |
|          | Helsinki: Street      | 140.4                   | 52.7         | 1.32                      | 13.0                  | 25 200                  |
|          | canyon (Summer)       |                         |              |                           |                       |                         |
|          | Helsinki: Highway     | 140.9                   | 56.7         | 0.54                      | 5.6                   | 9 000                   |
|          | Raahe: Industrial     | 147.7                   | 61.2         | 0.67                      | 6.6                   | 9 500                   |
|          | Helsinki: Harbour     | 169.2                   | <i>77</i> .9 | 0.96                      | 11.2                  | 8 700                   |
|          | Helsinki: Residential | 169.6                   | <i>57</i> .8 | 0.99                      | 9.3                   | 15 900                  |
|          | Raahe: Residential    | 205.5                   | 94.0         | 1.11                      | 11.8                  | 7 000                   |
| Central  | Prague: Highway       | 151.8                   | 50.7         | 2.71                      | 26.5                  | 55 900                  |
| Europe   | Düsseldorf: Highway   | 158.6                   | 70.4         | 1.41                      | 17.8                  | 41 400                  |
|          | Düsseldorf: Airport   | 183.6                   | 42.4         | 1.39                      | 18.7                  | 32 600                  |
|          | Düsseldorf: Urban     | 234.5                   | 47.4         | 1.52                      | 25.0                  | 16 600                  |
|          | Düsseldorf: River     | 251.7                   | 69.3         | 1.56                      | 28.6                  | 17 500                  |
|          | Prague: Urban         | 262.8                   | 80.1         | 1.37                      | 23.9                  | 11 700                  |
| India    | Delhi: Urban          | 278.1                   | 109.6        | 14.06                     | 256.9                 | 66 400                  |

In Fig. 6, the average CS contributed by ultrafine particles ( $CS_{0.1}$ ) is shown for all the studied environments. Also, the fraction of  $CS_{0.1}$  in total  $CS_{2.5}$  (CS contributed by 

Thus, only PN concentration is not enough to accurately estimate the condensation sink attributable to ultrafine particles even though the correlation was mainly good ( $R^2 > 0.6$  except highway sites in Helsinki, Tampere and Düsseldorf). The connection between PM<sub>2.5</sub> and CS<sub>2.5</sub> varied in the studied environments, slopes of the linear fits being 0.35–1.54 1/s/( $\mu$ gm<sup>-3</sup>) in Finland, 0.49–1.03 1/s/( $\mu$ gm<sup>-3</sup>) in Central Europe, and 0.48 1/s/( $\mu$ gm<sup>-3</sup>) in India (Fig S13–15). Also, the correlation was poor in road traffic and airport sites in Finland ( $R^2 < 0.5$ ). Hence, the connection between CS and traditional particle metrics (PN and PM<sub>2.5</sub>) can be strongly dependent on the environment and region.

Figure 6: CS attributable to ultrafine particles (CS<sub>0.1</sub>) in the studied environments. Also, the fraction of CS<sub>0.1</sub> in total CS<sub>2.5</sub> (of particles smaller than 2.5  $\mu$ m) are shown.

## 4 Discussion



The observed differences in the obtained CS size distributions (Fig. 5) indicate potential differences in both the aerosol health and climate effects of particles collected in different urban environments, and originating from different main sources. In principle, the CS size distribution describes the rate at which condensable vapours condense to different size ranges of the existing particle size distribution. A high value in the CS size distribution implies a higher transfer rate of these vapours from gas phase into particle phase at the certain particle size range. As different particle sizes have strongly varying efficiencies e.g., for lung deposition (e.g., ICRP 1994, Heusinkveld 2016), knowledge of the CS size distribution gives essential information of the existing particle population's potential for transferring condensable toxic compounds into the human








body. For example, the CS size distribution enables the determination of CS contributed by ultrafine particles (CS<sub>0.1</sub>) which deposit in the lung alveoli more efficiently than larger particles. It was observed that CS<sub>0.1</sub> concentration considerably varied in different locations (Fig. 6) depending on the nearby pollution sources. Also, one potential application for the information provided by the CS size distribution in terms of potential health effects could e.g. be the infiltration of outdoor emissions to indoor as particle filtration in building ventilation systems is size-dependent (Karjalainen et al. 2017). For example, the change of the particle size distribution after filtration could indicate varying health effects of the infiltrated or indoor emitted semi-volatile compounds.

Moreover, the observed results provide a valuable point-of-view when considering the health connections of the traditionally measured particle metrics such as PM<sub>2.5</sub> and PN concentrations. As seen in Table 3, Fig. 6 and Fig. S9-15, the linkage of CS, and especially CS<sub>0.1</sub>, with PM<sub>2.5</sub> clearly varies depending on the urban environment and geographic region. Lower PM<sub>2.5</sub> mass causes a shift of the CS size distribution to smaller particle sizes, i.e. to sizes that have more efficient deposition efficiency in the lung alveoli. Thus, it could even be possible that cleaner ambient air in respect of particulate mass concentration would lead to higher efficiency for semi-volatile compounds to reach e.g. the alveolar parts of lung, which could affect the health effects associated with PM2.5 concentration. For example, studies have reported that the PM2.5 doseresponse value seem to increase as the PM<sub>2.5</sub> decreases (e.g., Vodonos et al. 2018). Hence, relatively increased condensation of toxic vapours on ultrafine particles could be one partial explanation for the relatively increased health effects of PM<sub>2.5</sub> in areas with low pollution level. Also, as the connection between PN and CS (and CS<sub>0.1</sub>) varies depending on the urban environment (Fig. S10-12), the potential of ultrafine particle emissions to act as carriers of toxic vapours can vary depending on the location, plausibly affecting the variability of health effects associated with ultrafine particles. Furthermore, it's worth to note that most emitted semi-volatile compounds, e.g., from vehicles, condense to the particle phase already in the cooling process of initially hot aerosols (Ristimäki et al., 2007; Rönkkö and Timonen 2019). Therefore, the understanding of the CS size distribution of non-volatile primary emissions e.g., with different fuels or after-treatment processes could be valuable regarding the potential toxic effects of fresh exhaust that is a prevailing source of pollutants in urban city environments. In addition to the health effects, the observed differences in the CS size distribution are interesting in terms of the aerosol climate effects. For example, the vapour condensation on particles changes the coating of particles, thus, affecting how the particles interact with the incoming solar radiation (e.g. Riemer et al. 2019). Therefore, understanding the CS size distribution could help to estimate for which particle types and sizes condensation is the most important in certain location. Also, the activation of particles as CCN depends on particle size (50-100 nm, e.g., Kerminen et al. 2012). The differences in CS size distribution indicates that nanoparticle growth into sizes where they could act as CCN can depend on the environment and existing particle concentrations. Better characterization of CS size distribution could help to better understand CCN formation in different environments. In order to model the climatic impacts of particles accurately, it is important to characterize the factors affecting formation and change in the atmospheric particle size distribution in different conditions and environments. Accurate modelling of the particle size distribution on a global scale is notoriously difficult

and resource-consuming; adapting the CS size distribution in reporting observational data could be useful and rather simple





method to potentially provide additional constraining data for the important process of condensation, improving the accuracy of climate models.

To better understand the implications of the CS concentration and size distribution in terms of health and climate effects, it's also important to understand how well the current measurement methodologies enable CS size distribution measurement. In principle, CS measurement requires particle size distribution measurement. However, varying methods for particle size distribution measurement could affect the accuracy of CS size distribution measurement. The benefit of the ELPI+, the main instrument in this study, is that it covers the whole particle size range relevant for CS. Also, the diffusion charging of particles roughly correlates with CS due to the dependence of particle surface area. Hence, the ELPI+ enables a rather simple, but efficient method for CS size distribution measurement. On the other hand, the DMPS measurement also agreed well with the ELPI+ in terms of CS size distribution in this study, especially when considering the shape and mean size of the size distribution (Fig. 4). Therefore, according to the study, the current available size distribution methods seem to be well suitable for CS size distribution measurement, suggesting rather easy adaptation of current particle size distribution monitoring networks for CS size distribution measurement.

One interesting point-of-view is also the reasonably good performance of the ELPI+ single-factor CS calibration (Fig. 3), i.e., converting the diffusion charged current into CS concentration without considering particle size. In principle, the result indicates that also electrical particle sensors, like Partector (Fierz et al., 2014) and Pegasor PPS-M (Järvinen et al., 2015), that have become more popular in air quality monitoring, could measure CS with reasonable accuracy. Hence, an electrical particle sensor network could help to obtain CS information in various locations without the need for more complicated size distribution measurement devices. On the other hand, the sensors could not be utilised in the CS size distribution measurement, but they still could help to improve the skill of models focusing on condensation.

Still, it must be noted that there are several factors that influence the determination of the CS concentration and size distribution from experimental measurements that should be carefully considered. For example, in case of ELPI+, the effective density of particles affects the determination of the CS (see Fig. 4), and the absence of such information introduces a source of potential error, here approximated to be roughly  $\pm$  30 %. Also, the spherical particle assumption with mobility particle sizers (here DMPS) causes likely underestimation of CS: According to the results of this study, mobility particle sizer based measurement seem to report 20–30 % lower CS than diffusion-charger-based measurement. Therefore, it is not straightforward to conclude experimentally the most suitable and accurate method for CS measurement.

In addition to the mentioned uncertainties, another typical source of error in atmospheric CS measurements is the typical convention (Wiedensohler et al. 2012) of drying the ambient sample before performing size distribution measurements; this has the effect of reducing the size of hygroscopic particles which in turn can have a significant effect on the total condensation sink as well as the CS size distribution, as particle hygroscopicity is often size-dependent (Vu et al. 2015). This effect is difficult to quantify and require hygroscopicity and/or composition measurements. In cases where the ambient sample is dried either in a controlled manner, or e.g. due to dilution, the observed CS and CS size distribution should be considered a low-limit estimate, with the actual CS higher by a factor of up to 4 (Dal Maso et al., 2002). The ELPI+







measurements of atmospheric aerosols are typically not dried, and they do not have sheath flows, leading to observed distributions that are closer to ambient humidity than e.g. DMPS measurements. It should also be noted that the CS and CS size distribution reported typically (also here) are a theoretical quantity computed assuming that the condensing vapour is sulphuric acid with an accommodation coefficient of 1.0; in reality, the condensing compound of interest can have strongly different properties, and the actual condensation sink can be only a fraction of the theoretical one (see e.g. Tuovinen et al, 2021). These effects may also have implications to the CS size distribution which are difficult to quantify. Overall, it is fair to conclude that CS measurement is rather crude approximation of a highly complex atmospheric phenomenon, but, still, the results observed in this study suggest significant location dependent differences in vapour condensation, and the measurement of CS size distribution enables a rather easy method to better understand potential health and climate relevant differences in existing particle population regarding vapour condensation.

#### **5 Conclusions**

In this article, we explored the concept of the condensation sink size distribution. The concept is based on particle size distribution measurements, and it provides a tool which can be used to evaluate how the condensing gases and semi-volatile compounds are distributed as a function of particle size in gas-to-particle processes where the aerosol particles form a condensation sink for those compounds. We demonstrated the concept using ambient aerosol data measured by ELPI+ instrument, which provides the size distribution data with large particle size range, utilising diffusion charging of particles and particle size classification based on aerodynamic particle sizes. To do that, we determined particle size resolved conversion factors from electric currents to CS for the ELPI+ and investigated how the effective density of particles affect the conversion factors as well as how the ELPI+ measurement compares with mobility particle size measurement conducted with DMPS.

Our study shows the potential of CS size distribution measurement in terms of both aerosol climate and health effects studies as well as the suitability of current size distribution methods for CS size distribution measurement. The results show that CS size distribution can considerably vary in different urban environments and regions. In locations where PN concentration is high but PM<sub>2.5</sub> low, CS size distribution shifts to smaller particle size ranges (GMD<sub>CS</sub> < 100 nm), indicating more efficient lung deposition of condensable toxic vapour molecules as well as relatively increased growth of particles into the sizes where they could act as CCN. As PM<sub>2.5</sub> increases, the GMD<sub>CS</sub> seem to reach a plateau size around 300 nm (aerodynamic diameter), showing why the accumulation mode of particles typically peaks in the size range between 200–300 nm (aerodynamic diameter) regardless of the environment. Also, the varying CS size distributions show that the formation and changes of the atmospheric particle size distribution vary in different conditions and environments, which is important to understand when developing climate models. The comparison of the ELPI+ and DMPS measurement suggested that both methods seem suitable for CS size distribution measurement, and they agree reasonably well with each other. Overall,

adaptation of CS size distribution can potentially provide important additional information regarding aerosol health and climate effects with relatively simple measurement with respect to current measurement methodologies.

In general, the understanding of the gas-to-particle processes and the potential effects of condensable vapours on health and climate requires future research. For example, a large database of measured CS size distribution in different locations could help to identify the potential effects of CS size distribution on health and climate. Also, understanding of the condensation of other compounds than sulfuric acid, e.g., organic compounds, is essential for accurate measurements and analyses.

# Data availability

The utilised data are from publications <a href="https://doi.org/10.1016/j.envint.2023.108224">https://doi.org/10.5194/ar-2-271-2024</a> and <a href="https://doi.org/10.5194/ar-2-271-2024">https://doi.org/10.5194/ar-2-271-2024</a>

#### **Author contributions**

Teemu Lepistö: Conceptualization, Data curation, Formal analysis, Investigation, Methodology, Validation, Visualization, Writing (original draft preparation), Writing (review and editing)

Hilkka Timonen: Conceptualization, Funding acquisition, Methodology, Project administration, Writing (review and editing)
Topi Rönkkö: Conceptualization, Funding acquisition, Methodology, Project administration, Writing (review and editing)
Miikka Dal Maso: Conceptualization, Funding acquisition, Methodology, Project administration, Writing (review and editing)

#### **Competing interests**

The authors declare that they have no conflict of interest.

## Acknowledgements


This study is part of the AEROSURF-project, funded by the Research Council of Finland (grant no. 356752).

This work has received funding from the Horizon Europe research and innovation programme under grant agreement No 101096133 (PAREMPI: Particle emission prevention and impact: from real world emissions of traffic to secondary PM of urban air).

Also, ACCC-flagship (ACCC: Atmosphere and Climate Competence Center), funded by the Research Council of Finland (grant no. 337552, 337551) is acknowledged.

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
