# Peer review of "Size-resolved condensation sink as an approach to understand pathways how gaseous emissions affect health and climate"

_EGUsphere, 2025_

## Referee Comment (RC1)

Comments on "Size-resolved condensation sink as an approach to understand pathways how gaseous emissions affect health and climate" by Teemu Lepistö et al.

Teemu Lepistö et al. explores the concept of size-resolved condensation sink (CS size distribution) by calibrating an electrical low pressure impactor to measure CS and then utilize the method in urban aerosol measurements conducted in different areas in Finland, Germany, Czechia, and India, covering diverse sources (road traffic, airports, industrial sites). Results showed significant regional variations in CS size distributions. The study highlights the condensation sink (CS) size distribution in linking particle properties to health (e.g., toxic vapour transport) and climate (e.g., CCN activation) effects. It also notes limitations, such as uncertainties from particle effective density and simplified condensation assumptions, but concludes CS size distribution is a simple, effective tool for improving climate models and air quality assessments. The topic fits scope of ACP. The paper can be considered for publication after revisions that address the following concerns.

**Specific comments:**

- (1) Section 2.2.1 "Validation of the calibration" is the sole subsection under Section 2.2. Please revise its numbering to 2.3. Correspondingly, Section 2.3 "Measurement campaigns" should be renumbered to 2.4.
- (2) **Equation (3):** What are the values of  $D_{x,vap}$  (vapour diffusion coefficient) and  $M_x$  (molecular weight of the condensing vapour)? Based on **Lines 108–110**, I infer that the authors used values corresponding to sulfuric acid. So, was the condensation sink (CS) calculation in this paper solely focused on sulfuric acid? Whether this is the case or not, I recommend that the authors clarify this point explicitly.
- (3) Another question concerns why the authors chose sulfuric acid as the reference vapour rather than organic vapours—given that organic vapours may be more critical for condensational growth within the particle size range focused on in this study (>10 nm) (Riipinen et al., 2012; Sakurai et al., 2005)). I suggest including a discussion to address this choice.
- (4) **Section 2.2:** Many variables in this section are not clearly described, for instance,  $k_b$ . I recommend adding a nomenclature list that includes specific parameters, their corresponding definitions, and the values utilized in the calculations.

- (5) **Section 2.2:** In this section, what type of particle diameter is referred to (e.g., as mentioned in Line 105)—mobility diameter or aerodynamic diameter? Please clarify this explicitly in the related context.
- (6) **Table 1:** In Table 1, what does dpMeant in the "Stage" column refer to? Is it a mobility diameter converted from the aerodynamic diameter, or simply the aerodynamic diameter corresponding to each stage?
- (7) Line 236-238: The authors state in Figure 4 that "the relative difference between the instruments was clearly the highest with particles around 100 nm or smaller". However, it is difficult to identify the particle size corresponding to the peak relative difference from Figure 4 alone. In fact, I even observe that the peak of the relative difference in the third subplot (LRT) of Figure 4 appears to occur at approximately 200 nm. If the authors intend to illustrate this trend, it is recommended to supplement a particle size-resolved distribution plot of the relative differences to provide clearer evidence and possible.
- (8) **Line 273:** Why the elevated PM2.5 mass concentration relatively reduced the condensational growth of ultrafine particles. Please provide relevant discussions and supporting references.

(9) **Figure 2, 4 and 5:** Please number the subplots (e.g., (a), (b), (c)) consistently with Figure 1, and refine the caption.

**References**

Riipinen, I., Yli-Juuti, T., Pierce, J. R., Petäjä, T., Worsnop, D. R., Kulmala, M., & Donahue, N. M. (2012). The contribution of organics to atmospheric nanoparticle growth. Nature Geoscience, 5(7), 453-458. https://doi.org/10.1038/ngeo1499

Sakurai, H., Fink, M. A., McMurry, P. H., Mauldin, L., Moore, K. F., Smith, J. N., & Eisele, F. L. (2005). Hygroscopicity and volatility of 4-10 nm particles during summertime atmospheric nucleation events in urban Atlanta. Journal of Geophysical Research-Atmospheres, 110(D22). https://doi.org/10.1029/2005jd005918